# Emotional Health Detection in HAR: New Approach Using Ensemble SNN

**Luigi Bibbo' [1,*]** , **Francesco Cotroneo [2]** and **Marley Vellasco [3]**

1   Department of Information Infrastructure and Sustainable Energy,
   University Mediterranea of Reggio Calabria, Via dell'Università, 25, 89126 Reggio Calabria, Italy
2   Nophys S.r.l.s., Via Maddaloni 74, 00177 Roma, Italy
3   Department of Electrical Engineering, Pontifical Catholic University of Rio de Janeiro (PUC-Rio),
   Rua Marquês de São Vicente, 225, Rio de Janeiro 22451-000, Brazil
*   Correspondence: luigi.bibbo@unirc.it

**Abstract:** Computer recognition of human activity is an important area of research in computer vision. Human activity recognition (HAR) involves identifying human activities in real-life contexts and plays an important role in interpersonal interaction. Artificial intelligence usually identifies activities by analyzing data collected using different sources. These can be wearable sensors, MEMS devices embedded in smartphones, cameras, or CCTV systems. As part of HAR, computer vision technology can be applied to the recognition of the emotional state through facial expressions using facial positions such as the nose, eyes, and lips. Human facial expressions change with different health states. Our application is oriented toward the detection of the emotional health of subjects using a self-normalizing neural network (SNN) in cascade with an ensemble layer. We identify the subjects' emotional states through which the medical staff can derive useful indications of the patient's state of health.

**Keywords:** HAR; face emotion recognition; face detection; computer vision; deep learning; SNN; ensemble; vectorflow





## 1. Introduction

Human activity recognition (HAR) designates the complex of human action, which can be decomposed into human-to-human interaction, human-to-object events, and gestures [1]. Its objective is to detect data relating to the activities usually carried out by the elderly or those in need of care using sensors or observing a sequence of actions from videos or images. An applicable technique is facial expression recognition (FER) [2,3]. Human beings interact with each other through gestures and emotions [4]; therefore, facial expressions are a way to obtain emotional information [5] and can reflect a person's psychophysical state [6,7].

Thanks to various technological innovations, the emotion recognition and detection (EDR) has found widespread applications in different sectors and, according to some estimates, will have its highest growth rate in the coming years. During the pandemic, EDR technology was used by some companies to assess the state of satisfaction of workers who had been employed in smart working activities. Through computer vision algorithms, recognizing the emotions and moods of workers through their facial expressions, it was possible to assess the stress level to which they were subjected. Another sector in which it has found applications is learning and education. EDR technology has been used to assess the learning level of students by providing helpful guidance to educators to adopt the proper corrections to improve the learning process. Another sector in which EDR technology finds applications is healthcare. Thanks to the creation of smart homes and IoT technologies, patients can be provided with efficient healthcare without resorting to hospital admissions. A large amount of data can be acquired through computer vision and

sensors, which, when processed and analyzed, provide valuable indications to healthcare personnel and doctors to improve care and provide adequate services. This is aimed at improving the lifestyle of people in need of assistance.

The recognition of facial emotions is important because, from the analysis of the face, it is possible to detect the state of health of the subject, such as anxiety, depression, stress, or malaise, making a facial diagnosis possible. It is a beneficial technique in caring for the elderly; through the information provided, medical personnel can evaluate the type of intervention to reduce the state of the discomfort of the subjects. Some manifestations of the face can be associated with the first pathological symptoms facilitating the prevention of diseases that can degenerate. The eyes, in particular, can report physical and mental changes. This technique has ancient roots, in particular in traditional Chinese medicine (TCM), in which doctors performed a diagnosis of a disease by looking at facial features. They believed that pathological changes of internal organs reflected directly on the face allowed them to determine a diagnosis. Each anatomical part of the face reflected the state of functioning of each organ; for example, the cheeks were associated with the lungs, the lips with the digestive system and stomach, and the eyebrows with the nervous and respiratory systems. This technique, called "facial diagnosis," requires considerable experience in order to perform an accurate diagnosis. In modern times, the difficulty of obtaining a medical examination due to economic conditions and the scarce availability of medical resources for those who live in underdeveloped geographical areas has stimulated research to develop techniques and diagnoses using artificial intelligence. Finally, thanks to the technological evolution of deep learning, it has been found that computer-assisted facial diagnosis has low error rates [8].

While being a helpful tool, the use of deep learning in facial recognition presents some problems on which researchers work to improve efficiency. The main challenges must be addressed: accuracy, security, and privacy.

Accuracy is an essential aspect of the recognition process. The detected face may have lighting problems, poor image quality, low resolution, a blurred face, or different types of occlusions: systematic (hair, masks, clothes, and make-up) and temporary (hands covering the face or pose variations). These factors create errors in the faceprint feedback with faces in the database. Using a large amount of training data and 3D imagery improves accuracy.

Security refers to the risk of using facial recognition for identity theft or illicit purposes. Efficient ML and DL security systems already present in systems can provide adequate protection against illegal use.

Additionally, regarding privacy, the use of facial images must comply with the laws on processing personal data.

In interpreting images in the biomedical field, specific studies have been presented on improving the efficiency of deep learning. Below are some of the most significant examples.

Zhao et al. [9] developed an interesting method of deep learning in bioimaging called VoxelEmbed. Their work was finalized in the realization of an innovative multistream approach that facilitates embedding pixels with 3D contextual information. This solution arose from the need to meet the need for tools for the analysis of the dynamics of living cells. The segmentation and tracking of cell instances based on pixel incorporation have proven helpful for studying cellular dynamics. This method was validated through tests on four 3D datasets of the Cell Tracking Challenge.

Meanwhile, Zheng et al. [10] proposed a system to optimize the feature boundary of deep CNN through a two-step training method: a pre-training step and implicit regularization. Regularization is the process of enhancing the generalization ability of a CNN in order to train complex models while maintaining lower overfitting. It has a primary role in the tuning parameters of deep CNN. In the pre-training step, the authors trained the model to obtain the image representation for anomaly detection. Based on the anomaly detection results, the implicit regularization step re-trained the network to regularize the feature boundary and obtain the convergence. The implicit regularization can be considered as an implicit model ensemble. It can be regarded as the training process of a different network

that shares weights with different training samples; it can be viewed as a combination of a wide number of similar CNNs trained with different datasets.

Finally, Yao et al. [11] presented interesting findings relating to overcoming the problem of training self-supervised machine learning algorithms using massive data of biomedical images from databases or specialized journals. However, the images from such sources consist of a considerably large amount of compound figures with subplots.

In their work, these authors developed a framework (SimCFS) for separating compound figures using weak classification annotations. In particular, they proposed a system to separate the compound figures with lateral loss. The training stage contains these steps. SimCFS only requires single images from different categories. The pseudo compound figures are generated from the augmentation simulator (SimCFS-AUG). Then, a detection network (SimCFS-DET) is trained to perform compound figure separation. In the testing stage, they used only the trained SimCFS-DET for separating the images.

For evaluating the performance of different compound figure separation strategies, they used one compound figure dataset (called Glomeruli-2000) consisting of 917 training and 917 testing real figure plots from the American Journal of Kidney Diseases (AJKD).

The proposed method SimCFS-DET was then compared with the most used methods for separating compound figures showing better performance. ImageCLEF2016 was used as the dataset, containing 8397 figures, of which 6783 were for training and 1614 for testing. The system ultimately allows efficient distribution to new image classes.

In this work, a facial emotion recognition system based on an innovative approach of an SNN ensemble network is designed to support the work of medical staff in the diagnostic evaluation of the type of detectable malaise. The work after the introduction containing the importance of FER is articulated as follows: Section 2 presents the related works, Section 3 presents the analysis of the problem, Section 4 provides the methodology, Section 5 sets out the dataset and model design, Section 6 presents the results and discussion, and finally, Section 7 provides the conclusions.

## 2. Related Works

In recent years, many applications of emotion recognition have been developed using single modalities, multiple modalities, static images, and videos; here are some examples.

Jin et al. [12] proposed a system for identifying specific diseases using the deep transfer learning technique from facial recognition to facial diagnosis. With this technique, they solved the problem of the difficulty of finding images for facial diagnosis. They developed a suitable system for detecting and screening diseases using a small dataset. The study was aimed at identifying the following diseases:

- Thalassemia: a genetic blood disorder caused by abnormal hemoglobin production, which is a hereditary disease.
- Hyperthyroidism: an endocrine disease caused by excessive amounts of thyroid hormones T3 and T4.
- Down syndrome: a genetic disorder caused by trisomy of chromosome 21.
- Leprosy: an infectious disease also known as Hansen's disease, caused by the bacterium Mycobacterium leprae.

The methodology adopted was deep transfer learning (DTL), which transfers knowledge from a pre-trained deep neural network for facial verification and recognition. Starting from the recognition and verification of the face, the authors moved on to facial diagnosis.

Since these domains have the same feature space and related activities, it was possible to use a small dataset for deep transfer learning from facial recognition to facial diagnosis. The implementation of the model was based on MatLab's MatConvNet. The NVIDIA CUDA toolkit and CuDNN library were used for GPU acceleration.

The model was built through a fine-tuning phase of a pre-trained VGG-Face CNN and using pre-trained CNN as a feature extractor for the smaller dataset. The VGG-Face dataset contained 2.6 million images.

The model was tested on two cases of facial diagnosis. One was the detection of beta-thalassemia, which was a binary classification activity. The other was detecting four diseases, such as beta-thalassemia, hyperthyroidism, Down syndrome, and leprosy, a multi-class classification activity. Only 140 images from the dataset were used to detect a single disease, of which 70 were for facial images specific for beta-thalassemia and 70 for healthy subjects. Of the 70 of each type, 40 were used for training and 30 for testing. Comparing the results obtained with the VGG-Face model with those of traditional machine learning methods (AlexNet and ResNet50), it was found that its accuracy, being greater than 95%, is much better than that of the others. In further evaluating the algorithm, the authors performed a multi-class classification. In this case, 350 images from the dataset were used (70 for each face). Two hundred images are used for the training process (40 images for each face). One hundred and fifty images (30 images for each face) were used for the testing process. Although the classification process was more complex, the model had excellent accuracy for beta-thalassemia, Down syndrome, and leprosy. It had low accuracy for hyperthyroidism. The performance obtained was better than that of traditional machine learning methods. The accuracy was 93%.

Therefore, CNN as a feature extractor has been proven to be a suitable deep transfer learning method when using a small dataset for facial analysis.

Jin et al. [13] introduced an innovative deep-learning model to improve facial recognition. Many factors, such as lighting, variation of head pose, and lack of information about the spatial characteristics of the face, influence the accuracy of facial recognition. The latter can be overcome with the use of RGB-D sensors. In practice, images of RGB-D faces are challenging to find. Therefore, in the face of these color and depth problems connected to RGB sensors, these authors developed a model to obtain more accurate facial recognition based on deep learning. The proposed solution allowed for obtaining depth maps from images of 2D faces in place of those obtainable with a depth sensor. In particular, they designed a neural network model called "D+GAN" (Depth plus Generative Adversarial network), with which they performed multi-conditional translation from image to image with facial attributes. Compared to the normal two-component GANs generator and discriminator, this network had the advantage of generating high-quality face depth maps by making greater use of facial attribute information and determining sex, age, and race categories. The facial recognition process was divided into the following phases:

- Capturing RGB face images.
- Image preprocessing to remove the image background from the face evenly.
- Generating face depth maps
- Merging images using the unsampled Shearlet transform (NSST).
- Using different tools for face recognition.

The Bosphorus 3D Face Database and the CASUA 3D Face Database were used to train the model, and the BU-3DFe Database was used for testing.

To evaluate the quality of the depth maps obtained, face depth maps were created for each of the previous datasets using different techniques:

➢ Monodepth2
➢ DenseDepth Method (KTTI)
➢ DenseDepth (NYUDepth)
➢ DenseDepth (NYUDepthV2)
➢ 3DMorphable Model (3DMMI)
➢ Pix2Pix
➢ CycleGAN

and compared with D+GAN.

Numerous experiments were carried out to validate the model. The authors compared the eight depth maps obtained with the techniques cited for each dataset used. The results showed that the outputs generated by D+GAN show more detailed depth information for

all three datasets. Even in the correlation graphs of the various output images, there was a higher quality for the depth maps generated by 3DMM, Pix2Pix, Cycle GAN, and D+GAN.

For a quantitative analysis of the face depth maps generated by the different models, SSIM, RMSE (root-mean-squared error), and PSNR (peak-signal-to-noise ratio) were used. SSIM is a parameter used for assessing the structural similarity of images. It is used to evaluate the quality of the processed image compared to the reference one. Additionally, from a quantitative point of view, these three indices for D+GAN depth maps were the best.

The four models, PCA, ICA, Facenet, and InsightFace, were used as face recognition methods. ENTL, Yale, UMIST, AR, and ERET were used as datasets. From the experiments, it was found that for each dataset of the five used, the best results were obtained when operating with the pseudo-RGB-D facial recognition modes compared to the RGB mode. The best result was obtained when combining the FaceNet model with the ORL database.

Ghosh et al. [14] developed an innovative model of recognition of human emotions using analyses of both physiological and textual characteristics considering heart rate [15] and blood pressure, or changes in pupil size or even textual analysis. Their study analyzed five emotions: anger, sadness, joy, disgust, and fear. The model was built taking into account two physiological characteristics: facial muscle movements and HRV combined with textual analysis. Heart rate variability (HRV) is the oscillation of heart rate over a series of consecutive heartbeats. This study's novelty lay in the analysis of the combination of these characteristics with the classification of emotions obtained by applying a deep learning model based on RNN (Recurrent Neural Network). The methodological approach used to collect the characteristics corresponding to each of the five classes of emotions mentioned above involved showing 500 subjects belonging to different cultural backgrounds and age groups a series of photograms of various films to arouse emotions, which were the object of the study. The data on HRV were captured using a wearable device containing various biometric sensors such as a heart rate monitor, blood pressure measurement sensor, and body temperature sensor. The parameters measured were heart rate, systolic blood pressure, and diastolic blood pressure concerning the above categories of emotions.

Sixty-eight reference points have been identified on facial images to extract features from facial muscle movement. The positions of these points vary according to different emotions. The characteristics associated with the different landmarks of the face make it possible to distinguish the five different emotions. Only 47 landmarks were used in this study. Considering that each facial reference point is identified by x and y coordinates, a 94-dimensional characteristic vector was derived for each emotional category for each person. For textual analysis, ISEAR public datasets were used [16]. This dataset consists of several blogs written by different subjects in a specific emotional state. Each blog is associated with a label for emotional class. The datasets were associated with the five emotions: anger, sadness, joy, disgust, and fear. The ISEAR dataset consists of 7666 samples, but 1400 samples were used for each of the five emotional categories above. Using the word-emotion lexicon of the National Research Council (NRC), Canada, tokens related to each emotional category were extracted. Features were studied using long short-term memory (LSTM) and bidirectional long short-term memory (BLSTM) variants of the RNN classifier. The experimental results showed a classification accuracy of 98.5% in the case of BLSTM and 95.1% for LSTM.

Dahua et al. [17], for the recognition of emotions, used a multimodal model that, compared to the single-mode model, used complementary information that improved classification accuracy. They merged the features extracted from electroencephalography (EEG) to detect emotions continuously with those derived from facial expressions. To arouse emotions in the subjects, films and excerpts from the SEED dataset (SJTU Emotion EEG Dataset) [18] representing types of negative, neutral, and positive emotions were screened, and simultaneously EEG signals and facial expressions were recorded separately. Ten subjects participated in the experiment. Six sessions of movie clips representing a combination of the three classes of emotions were presented to each subject. Before each visualization session, the subjects were required to stay relaxed for 10 s to obtain the baseline

to capture the change in emotion. The length of each clip is about 200 s. A 30 s break was provided between one film and another. Ten observers, specialists in psychology, were employed to carry out the continuous annotation of the response of the facial expression of the subjects. For this phase, the DARMA program was used, which allowed continuous evaluations of valence and excitement to be collected when viewing audio and video files.

With the help of a joystick, the observers performed the continuous annotation of emotions. The EEG data were sampled ussing the Emotive EPOC headset equipped with 14 acquisition channels. The ECG signals were preprocessed to delete artifacts by applying the 4–47 Hz bandpass filter and the spatial filter based on independent component analysis (ICA). The PSD (PhotoShop Document) features were extracted using the short-term Fourier transform (STFT). The PSD features are correlated with emotions in different bands, such as theta bands (4–7 Hz), alpha bands (8–12 Hz), beta bands (13–30 Hz), and gamma bands (30–47 Hz). The 56 features (14 channels × 4 frequency bands) were used to represent the EEG signals. After this phase, feature selection was performed to simplify the model and to improve performance by reducing irrelevant or redundant features. The authors applied t-distributed stochastic neighbor embedding (t-SNE), a nonlinear feature selection algorithm that was very efficient in computer vision. The valence predictions of EEG were obtained by support vector regression (SVR). Facial geometric features were extracted using the facial reference point localization model for facial expression. In particular, the inclination of the forehead, the extension of the opening of the eyes, the extension of the mouth, and the inclination of a corner of the mouth were chosen as facial features. These features were extracted by considering the coordinates of 29 landmarks in the eye and mouth. SVR was also applied for facial predictions. Both features were merged. Long short-term memory networks (LSTM) were utilized to accomplish the decision-level fusion and capture the temporal dynamics of emotions. To verify the validity of their classification method, the authors compared their t-SNE feature selection method with principal component analysis (PCA) with different dimensions. The results showed that the precision of EEG-based emotion recognition improves with decreasing feature dimensions. Both methods demonstrated the validity of the recognition system.

Moreover, t-SNE achieved more significant improvement than PCA. The best performances achieved by t-SNE and PCA were 0.534 ± 0.028 and 0.464 ± 0.032, respectively, when the dimension of the mapped feature was 15. The results of continuous emotion recognition showed that the fusion of two modalities provided better results than EEG and facial expressions separately. For the results of a single modality, facial expressions were better than EEG. The experimental results found that three steps of LSTM yielded the best CCC (concordance correlation coefficient) of 0.625 ± 0.029.

The possibility of using wireless devices and networks has stimulated research in the design of innovative models capable of recognizing human activities through the collection of physiological, environmental, and position data to obtain valuable information on the state of health of people, to develop intervention strategies to improve living conditions. In addition, the development of machine learning algorithms makes it possible to deduce human emotions from sensory data that can facilitate the identification of mental situations in need of help.

In this context, Kanjo et al. [19] proposed a model that is based on the detection of emotions in motion and in real-life environments, different from other models in which emotions are detected in laboratory environments and with samples in which emotions are stimulated by audiovisual means or by asking participants to perform activities designed to induce emotional states. The experience was developed with the following:

- The use of multimodal sensors: physiological, environmental, and position data collected in a global template representing the signal's dynamics together with the temporal relationships of each mode.
- The application of different deep learning models to extract emotions automatically.
- Collecting data in real situations from subjects wearing a bracelet and a smartphone.

- In classifying emotions, the characteristics of the three types of signals are examined individually and combined.

The authors used the EnvBodySens dataset, already tested in a previous work [20], which collected data from 40 participants who walked on specific paths. The data were obtained for heart rate (HR), galvanic skin response (SGR), body temperature, motion data (accelerometer and gyroscope), environmental data such as noise levels, UV, atmospheric pressure and location data, GPS locations, and self-reported emotion levels recorded on Android phones (Nexus), wirelessly connected to Microsoft Wrist Band 2 [21]. The self-reported data referred to the responses given by the participants about the sensations they experienced while walking based on a predefined scale of emotions. The data collected included 550,432 sensor data frames and 5345 self-report responses. The signals were preprocessed and subsequently inputted into a hybrid model of a convolutional neural network and long short-term memory recurrent neural network (CNN-LSTM). The results showed that deep learning algorithms effectively classify human emotions when using many sensor inputs. The average accuracy was 95%. In addition, tests carried out using MLP, CNN, and CNN-LSTM models showed that with the hybrid model, the accuracy of emotional states increased by more than 20% compared to a traditional MLP model.

The application developed by Suraj et al. [22] was part of the research aimed at the automatic detection of emotions, using deep learning algorithms, to detect pain or discomfort to help medical personnel immediately activate the most suitable treatments. The solution adopted is based on the use of a CNN network to which images of the face and mouth are transferred. The authors created the model through human face detection, eliminating unwanted components using a webcam. Images underwent a preprocessing step to convert from RGB to grayscale using OpenCV libraries. Histogram equalization was performed to unify and improve image contrast for better edge identification. Next, a cascade Haar classifier was used to recognize the mouth and eyes in each frame. The classification of emotions was performed by the last level of the CNN network (SoftMax).

Experimental results showed that this system can detect normal emotions, pain, and fatigue accuracy of 79.71%.

## 3. Emotion Recognition Analysis

To offer readers a clear understanding of how emotions can be derived through visual images of the face, we developed an analysis of the theme reported below.

### 3.1. Detection Technique

The FER is also used in human intention prediction (HIP), which represents an emerging area of research in which the system, through the collected data, predicts human behavior to improve assisted living.

Face recognition in the context of the HAR in healthcare can be applied for:

- Detecting neurodegenerative disorders;
- Detecting states of depression or, in general, identifying subjects who need assistance;
- Observing the condition of patients during medical treatment;
- Detecting psychotic disorders;
- Monitoring anxiety states;
- Detecting pain or stress.

Human behavior can be characterized by analyzing facial expressions through the vision system. The face is the most expressive part of our body; it makes visible every emotional trace, thus making a face the primary source from which to obtain information on emotions.

Abdulsalam et al. analyzed how emotions can be detected [23]. They can be recognized through unimodal social behaviors, such as speech, facial expressions, texts, or gestures; bimodal behaviors, such as speech associated with facial expressions; or multimodal behaviors, such as audio, video, or physiological signals.

As part of our study, we focused on recognizing emotions through facial expression without considering other methodologies that use voice, body movements, or physiological signals.

Ekman was one of the first researchers to study emotions and their relationship with facial expressions; his research demonstrated the universality and discretion of emotions following Darwinian theory [24]. Over the years, he developed a set for the recognition of emotions based on a series of stimuli called POFA (Pictures of Facial Affect) consisting of 110 black and white images. According to Ekman et al. [25], in nature, there are two different categories of emotions, primary or universal emotions, and secondary or complex emotions. The former can also be present in other animals, as Charles Darwin argued in *The Expression of Emotion in Man and Animals*; the latter, however, is present only in human beings. Primary emotions are universal and innate emotions.

On the other hand, secondary emotions are affected by environmental and socio-cultural influences. The primary emotions are also essential: anger, fear, sadness, happiness, disgust, and surprise. Complex emotions include joy, envy, shame, anxiety, boredom, resignation, jealousy, hope, forgiveness, offense, nostalgia, remorse, disappointment, and relief.

In 1992, Ekman expanded his list of basic emotions, adding contempt, embarrassment, guilt, and shame to those already known. For each of the primary emotions, there are characteristic elements of the face that allow one to identify the type of emotions:

Anger, generated by frustration, manifests itself through aggressiveness and can be identified through the following features: a flushed face, hard look, dilated nostrils, clenched jaws, lowered eyebrows, and tight lips. Some of these characteristics are also present in expressions of fear. However, the eyebrows, the forehead movement, and the type of mouth allow us to differentiate the two expressions. In fear, the eyebrows are raised, the eyelids are stretched, and the mouth is open.

Happiness, the manifestation of a mood of satisfaction, is one of the easiest emotions to recognize because a smile appears on the person's face. The lips can be joined in the smile or open, including the teeth and cheeks raised. The more pronounced the smile, the more the cheeks rise.

Disgust is a feeling of repulsion, and its characteristics are: clenched nostrils, raised upper lip, and curled nose. The greater the sense of disgust, the more pronounced the upper lip and the wrinkling of the nose will be.

Sadness is identified through the forehead and eyebrows: the first displays a frown, and for the second, the inner corners are raised. Sometimes sadness can be confused with the emotion of fear.

Surprise, the manifestation of the state of mind in the face of an unexpected event, is a type of brief emotion in which the eyebrows appear curved and raised, the eyes wide, the forehead wrinkled, and the jaw is lowered, causing the lips to open.

Fear, an emotion produced in the face of a dangerous situation, is identified through the following characteristics: raised eyebrows, wide-open eyes, dilated pupils, joined eyebrows, and elongated lips.

There are two methods for studying facial expression: one based on an analytical method through which the mimic components that contribute to the determination of a specific facial expression are identified and the other based on the judgment of the facial expressions manifested. The facial action coding system (FACS) can be used as an analytical method. This system, developed by Ekman and Friesen, constitutes a measurement system to evaluate the movements of facial expressions of emotion. It can be used to identify the internal and emotional state of the subject. Through the analysis of facial micro expressions, which are involuntary and rapid expressions, it is possible to deduce indications of hidden thoughts and emotions of the subject. They appear with a fraction of a second and reveal the subject's genuine emotion.

The system is based on the coding of evaluators based on the presence and extent of facial micromovements, called facial action units (AU), such as the face, eye, and head movements. Face emotion recognition is a technology that belongs to the field of "Affective computing" [26], enabling automatic systems to interpret and recognize human emotions.

It is an interdisciplinary field that exploits computer science, psychology, neuroscience, and cognitive science. It is a technology that reasonably and accurately recognizes emotions from visual, textual, and auditory sources. High-resolution cameras and powerful machine learning capabilities allow artificial intelligence to identify emotion through facial expressions. It is used in various fields of application, as already seen above, including health, to study stress and psychophysical disorders.

### 3.2. FER Structure

Face emotion recognition (FER) is a technique for the recognition of emotions through the analysis of facial expressions in multimodal form.

FER has increased in the field of perceptual and cognitive sciences and affective computing with the development of artificial intelligence techniques, virtual reality [27], and augmented reality [28,29]. Different inputs are available for the FER, such as electromyography (EMG), electrocardiograms (ECG), electroencephalograms (EEG), and the a camera; the latter is preferable because it provides more information and does not require the use of wearable devices. The technology on which the FER is based uses mathematical algorithms to analyze faces acquired from images or videos to recognize emotions or behaviors through facial features. Recognition systems can use 2D images as input data, but newer approaches employ 3D models or combined 2D–3D models called FER multimodal models [30]. Three-dimensional technology performs better, but due to the high resolution and frame rate, it requires more computational power as the amount of data captured in 3D databases increases.

In addition to traditional approaches [31], deep learning-based algorithms can be applied for extraction, classification, and recognition activities.

FER is divided into the following phases: image acquisition, image processing, face detection, feature extraction, and emotion classification (Figure 1).

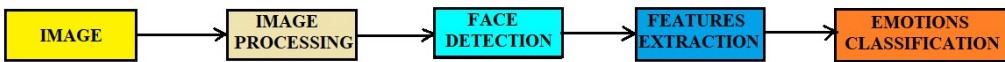

**Figure 1.** Flow chart of facial emotion recognition.

➢ Image processing is a preliminary phase to eliminate all interfering factors in the input image that can affect classification performance and complicate processing. It consists of locating and extracting the region of the face. It is used to eliminate the background noise through processing filters and to normalize the image's color. For example, one of the most commonly used filters to obtain a sharper image is RIR (Regularized Inverse Auto-Regressive) [32].

➢ Face detection involves distinguishing faces in an image or video and constructing bounding boxes for faces. One algorithm used for this purpose is the Viola–Jones algorithm, developed initially for object detection. This algorithm examines the minor features of a human face in an image, and if all these features are found, the algorithm predicts that there is a face in that image or a secondary image. Its application requires that requirements such as full-view, frontal, vertical, well-lit, and life-size faces in fixed-resolution images are met. Paul Viola and Michael Jones modified Haar's wavelets to develop so-called Haar-like features. A Haar-like feature considers adjacent rectangular regions in a sensing window, adds pixel intensities in each region and calculates the difference between these sums. This difference is used to categorize subsections of an image.

➢ Feature extraction is the process of extracting reference points that facilitate the algorithm to recognize the expression. Extraction methods can be different depending on the type of input image. For static images, the extraction method can be based on "geometric features" or "aspects". One commonly used geometric feature model is the active shape model (ASM) [33]. It consists of creating a suggested shape by looking at the image around each point for a better location for the point. Based on this aspect, local feature analysis (LFA) methods use the entire face or specific

measures to extract facial changes. The commonly used methods are local binary pattern (LBP) and Gabor feature extraction. LBP [34] is a texture operator defined as an ordered binary sequence of color depth comparisons between pixel p and pixels belonging to the neighborhood under consideration. To calculate the LPB code, for each generic pixel "p", the 8 "x" neighbors of the center pixel are compared with the pixel p and assigned a value of one if $x \geq p$. Calculating LBP on the entire image means producing a feature vector consisting of a histogram as an output result.

➢ Gabor feature extraction applies a series of filters to extract features. They are extracted from sequences of dynamic images and are derived from changes in expressions and the displacement of characteristic points of the face [35].

➢ Emotion classification has the task of identifying which emotions correspond to the facial features examined. The following is an overview of the approaches based on traditional methods and those based on deep learning [36]. In facial recognition technology, data separation is crucial, belonging to the same class. A class represents all data from the same subject. Linear discriminant analysis (LDA) and principal component analysis (PCA) are among the most commonly used classifiers. Both aim to separate data into classes. LDA [37] is a method that transforms image vectors into a low-dimensional space maximizing data separation between classes and minimizing dispersion within the classroom. That is, it groups the images of the same class and separates the images of a different class. LDA allows for identifying the aim of an objective evaluation of the visual information present in the features. Similarly, PCA [38] is an algorithm that transforms image vectors by reducing large dimensions into smaller values while preserving as much information as possible and separating data into the classroom. It employs eigenvalues and eigenvectors to reduce dimensionality and projects data samples onto a small space.

Another algorithm used for classification is the K-nearest neighbor algorithm [39]. Classification is performed by comparing the sample with its neighbors. The input consists of the training samples, while the output is the result of the sample belonging to a class. The sample is classified through the class to which its neighbors belong. Based on the value of **K**, the **K** elements closest to the sample to be examined are considered. Based on most elements of a given class, the sample under examination will be assigned to the same class.

Among the algorithms based on supervised learning models are support vector machines (SVMs) [40], which are binary linear classification methods. With the two training datasets, each identified according to the class it belongs to between the two possible classes, the model assigns the new examples to one of them. The algorithm is based on the identification of the separation line between the two classes that maximizes the margin between the classes themselves, where the margin means the minimum distance from the line to the points of the two classes. The so-called support vectors achieve this goal using only a minimal part of the training dataset. Supporting vectors are those datasets that reside on the margin and are used to perform classification.

Finally, random forest is a classifier formed by combining decision trees [41]. The algorithm builds multiple trees based on randomly selected subsets of the training dataset, then aggregates the predictions of each tree to choose the best prediction. Random forest belongs to the class of algorithms called "Ensemble". They work based on a combination of machine learning algorithms to create a predictive model that ensures better performance. There are different methods of aggregation, including bagging and boosting. Bagging consists of sampling the initial dataset N times, training it N times, and choosing the category most frequently for classification. On the other hand, boosting is based on a sequential process in which, at each step, the previous model is improved by correcting errors; each model depends on the previous one and tends to decrease the error.

Deep learning-based FER approaches significantly reduce reliance on models based on physical facial features and other preprocessing techniques, enabling learning directly from input images. The convolutional neural network (CNN) and recurrent neural network (RNN) are the most widely used network models.

### 3.3. Neural Network

The CNN network is a deep neural network that learns the characteristics of data layer by layer through a nonlinear structure. It consists of a set of several layers that have the function of extracting the characteristics of the input images and a completely connected terminal layer that acts as a classifier. They are suitable for analyzing images in specific datasets and classifying objects within them. Each processing layer contains a convolutional filter, a trigger function (Relu), and a pooling function. At the end of each processing step, an input is generated for the next level. The convolution subjects the images to a series of filters, each of which manages to activate specific characteristics of the images to create the feature map that becomes the input for the next filter. The activation function is intended to introduce a non-linearity into the system using nonlinear functions, and it can cancel negative values obtained in the previous classes. The pooling function obtains images with a particular input resolution. It returns the same number of images with fewer pixels, thus reducing the size of the output matrices and the number of parameters that must be learned from the network. At the end of the convolutional layers is the fully connected level (FC), which aims to identify the classes obtained in the previous levels according to a certain probability. These operations are repeated on multiple levels, and each level learns to classify different characteristics. A fully connected level and classification level are used to provide classification output. Designing a CNN network requires a training phase followed by a testing phase. During the training phase, the images are labeled and transferred to subsequent levels to allow conversion from the original input representation layer to a higher-level and more abstract representation to build the reference feature maps with which the network must compare the output feature maps. Each class represents a possible answer that the system will choose. During the recognition phase, the network follows a classification operation to identify which class the input image belongs to, identifying the one with the highest probability.

RNN is a feed-forward neural network similar to the CNN network. It still has an input layer, hidden intermediate levels, and an output layer. Land connections between nodes form a graph directed along a timeline. While in CNN, neurons of the same level cannot communicate with each other but can only send signals to the next layer, in RNN, neurons can also admit loops. They can be interconnected even to neurons of an earlier level. These networks link backward or to the same level. They can use their internal memory to process any input sequence. The output of a neuron can influence itself in a subsequent time step or affect the neurons of the previous chain, which will interfere with the behavior of the neuron on which the loop closes. RNN networks can process a data timeline, unlike classical feed-forward networks where the data provided are static. A timeline can be thought of as a function sampled over several moments.

Deep learning methods must use extensive datasets to achieve a high recognition rate, and so the algorithms do not work well if a few subjects form the datasets.

### 3.4. Dataset Used

The public databases used for analyzing emotions are the BU-3DFE and the BU-4DFE.

The BU-3DFE [42] is a 3D facial model database at Binghamton University containing facial images of 100 subjects of different ethnic and racial origins. Each subject was scanned with seven expressions. Except for the neutral expression, each of the six basic expressions (happiness, disgust, fear, anger, surprise, and sadness) includes four intensity levels. Thus, there are a total of 2500 3D facial expression models. Each expression pattern has an image of the corresponding facial texture captured at two views (approximately $+45°$ and $-45°$). As a result, the database consists of 2500 two-view texture images and 2500 geometric shape models. To analyze facial behavior from static 3D to dynamic 3D space, the BU-3DFE was extended to the BU-4DFE [42]. The new database of high-resolution 3D dynamic facial expressions refers to 101 subjects of different sexes, ages, racial and ethnic origins. Three-dimensional facial expressions were captured at a rate of 25 frames per second. Each sequence of expressions contains approximately 100 frames. Each

subject performed the six basic emotions, ending with the neutral expression. The database contains 606 sequences of 3D facial expressions.

Another database used is the Bosphorus [43], which contains 2D and 3D images of 105 subjects, of which a third are professional actors and actresses. The data were collected in the laboratory, and the subjects were instructed to perform the seven basic facial expressions. The scans for the 105 subjects were carried out considering different poses, expressions, and occlusion conditions. The total number of facial scans is 4666. This database contains examples of the unit of action (A.U.) faces defined in the facial action coding system.

Other databases with visual sequences and images are available for studying emotions; some examples are presented below.

The extended Cohn–Kanade Dataset (cK+) [44] contains 593 sequences of 123 subjects. The sequence of images varies in duration from 10 to 60 frames. The images are labeled with seven emotions, including six basic emotions and contempt. All images were taken with a constant background. To avoid mistakes during the training phase, the labels were assigned respecting the coding of FACS emotions.

Another database is the CASME [45], which contains spontaneous microexpressions. Microexpressions are fleeting facial expressions that reveal authentic emotions that people try to hide. From the 1500 facial movements filmed at 60 fps, 195 microexpressions were selected. Samples from the dataset were taken from thirty-five participants. Each clip has a minimum length of 500 ms. The images were labeled based on psychological studies and participants' self-assessments.

Still, the FER-2013 [46] is a widely used dataset containing 28,000 training data, 3500 validation data, and 3500 test data. Land images are stored in a spreadsheet where the pixel values of each image are reported in cells per row. The images were obtained using Google search and then grouped by emotional classes. The images were collected from varying poses, ages, and occlusion.

BAUM [47] is a spontaneous audiovisual facial database of affective and mental states. Video clips were obtained by shooting subjects from a front view using a stereo camera and a semi-profile view using a mono camera. Subjects were shown images and short video clips to evoke emotions and mental states. The target emotions are happiness, anger, sadness, disgust, fear, surprise, boredom, and contempt; mental targets are uncertain (even confused, indecisive), thoughtful, focused, interested (even curious), and annoyed. The database contains 273 clips obtained from 31 subjects (13 females, 18 males) with the age range of subjects being 19–65 years.

## 4. Methodology

A deep learning architecture is certainly not an innovative approach; in recent years, numerous applications have been developed and shown excellent facial and emotional recognition results. A deep neural architecture represents a practical solution; it allows one to analyze and extract the characteristics of each face to train the network and associate the characteristics of a new image with one of the seven emotions.

The innovation brought by our project is the development of an AI classifier based on a set of classifying neural networks whose outputs are directed to an ensemble layer [48]. In particular, the networks are self-normalizing neural networks (SNN) [49]. We can operate assumptions for which the problem is framed in the detection of phenomenological expressions that:

- They are finite and contained.
- They may have a certain degree of overlap in their manifestation.
- They evolve in a specific time frame, during which common randomness mechanisms govern the type of variation; in other words, the comparison between different temporal evolutions of the same phenomenology has little variability.

- The last stage of this evolution represents more distance between the expected classes (or with less overlap between classes), which can be defined as a maximum characterization event.

The architecture consists of 6 SNNs, each trained to identify the six emotions. The networks are cascaded, and each is dedicated to detecting the presence or absence in the input image of a single specific emotion (among the six present in this study) assigned and associated with it. Each neural network is trained with its images for a specific emotion. Each network will produce two outputs, of which the first identified with EM, through a numerical enhancement (from 0 to 1), will confirm the correspondence of the detected emotion with that assigned to the network, and the second identified with AM, similarly through a numerical enhancement (from 0 to 1), will signal the presence of another emotion than that assigned to the specific network. If, for example, the first network has been trained to detect anger, the eligible cases will be EM1t1 = 1 and AM1t1 = 0 if the emotion is anger, and EM1t1 = 0 AM1t1 = 1 when the emotion detected is "another" different from anger.

These outputs are then transferred to the ensemble layer, which provides an accurate result by analyzing the outputs of the individual networks according to statistical logic. Ensemble is an algorithm that combines several trained models, allowing them to obtain better predictive results than single models.

Wanting to apply this architecture to the time interval during which the face passes from a "resting" stage to the stage "of complete characterization" of a specific emotion, it will be necessary to provide as an input to neural networks three frames obtained from a single specific video. In this case, the ensemble classifier will consist of 18 neural networks, six dedicated to identifying the six emotions for each video frame. Its architecture is represented by the diagram shown in Figure 2.

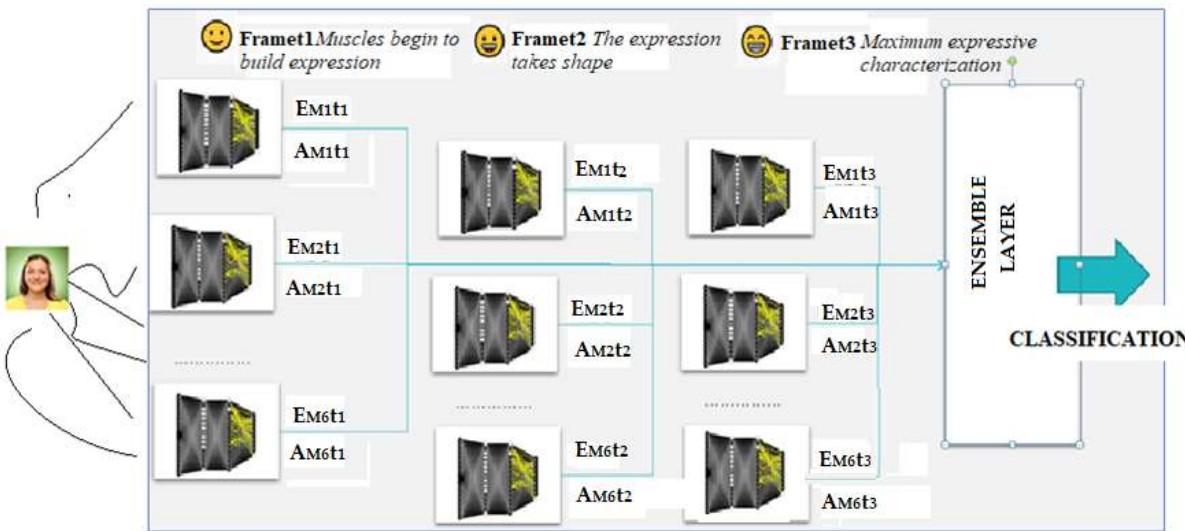

**Figure 2.** Network architecture.

The system's functioning can be described by assuming that we want to classify a specific emotion that we already know, such as "happiness." With its three frames, this emotion is inputted into the system. Assuming that the M2 network has been trained for this type of emotion, we will obtain that, for Frame 1, EM2t1 will take on a more significant weight value than all the EMit1 of the other five networks. Similarly, when we move on to analyze Frame 2, in which the expression takes shape, the EM2t2 will increase its weight value, resulting in more significance than the EMit2. Finally, with frame three, we will obtain a maximum evaluation of EM2t3 far superior to the other values for EMit3. Finally, the Classification Ensemble module, in analyzing all of the EMitj, determines as a predictive value the classification obtained from EM2t3, considering the temporal variation obtained.

This mode of analysis performed at three different time intervals allows us to evaluate the difference, even on the same individual, between the movements that the face performs during the moments leading to the full manifestation of emotion. It allows us to evaluate whether it is the result of a subconscious reaction to an event ("genuine reaction") or, vice versa, produced by an act of conscious voluntariness ("voluntary fiction").

This approach is challenging to implement since building a training set with the visual traits described above is complex. The extraction from the videos of the three frames that belong to the exact configuration of the muscles of the face relative to the evolution of a specific emotion needs to be revised. For example, extracting frames with only time synchronization in mind could compromise model inference. While the evolution of expressions can be correlated, the timing of expressions can vary significantly from individual to individual.

Therefore, finding such videos at a helpful quantity for training networks is challenging. The hypothesized architecture remains valid, excluding the temporal component, considering that a single image is instantly acquired. Even with this mode, the qualities related to the performance and near observability of inferential states remain preserved. For these reasons, a simpler architecture was applied in this work, as shown in Figure 3.

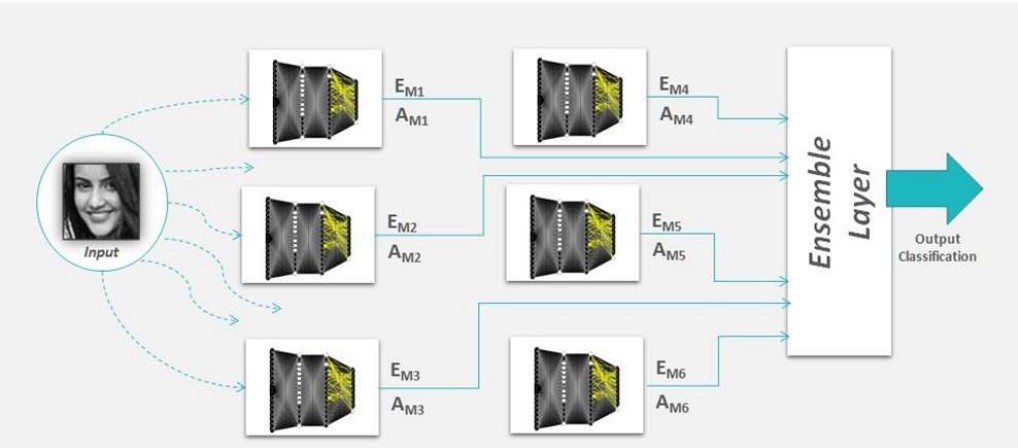

**Figure 3.** "Single frame" Configuration.

The input images in this configuration can be assimilated to the frame at time t3 of the configuration in Figure 2. The images are supposed to be no longer sequences of movie frames. Instead, the images of the training set are all referable to the stage of the maximum characterization of emotions.

Compared to a single neural network, this configuration has the advantage of implementing a classifier with inferential states that are "almost observable". With this solution, we can overcome the problem that afflicts a specific classifier if it produces an erroneous result. This classifier provides the advantage of being able to intervene when it is affected by development or model errors in the case of unwanted outputs. In the case of using neural networks, it is impossible to identify the "neuron" to be replaced as it is the entire network that has extracted a model that does not conform to the phenomenology to which it was applied. So, with an AI Ensemble classifier, it becomes possible to identify the subnet that gave the output altering the correct functioning of the ensemble module. It is possible to intervene directly in that specific subnet. This peculiarity, in addition to providing elements to understand why and how the error was born, allows for obtaining improvements in all parts of the classifier.

## 5. Dataset and Model Design

The research activities focused mainly on identifying, studying, and highlighting how the proposed AI ensemble approach can provide different advantages in the FER field. With this in mind, design choices can be characterized as "stress tests."

In particular, a Kaggle dataset was chosen for the training and test sets (Figure 4). The dataset also predicts the "neutral" emotion, which we did not consider.

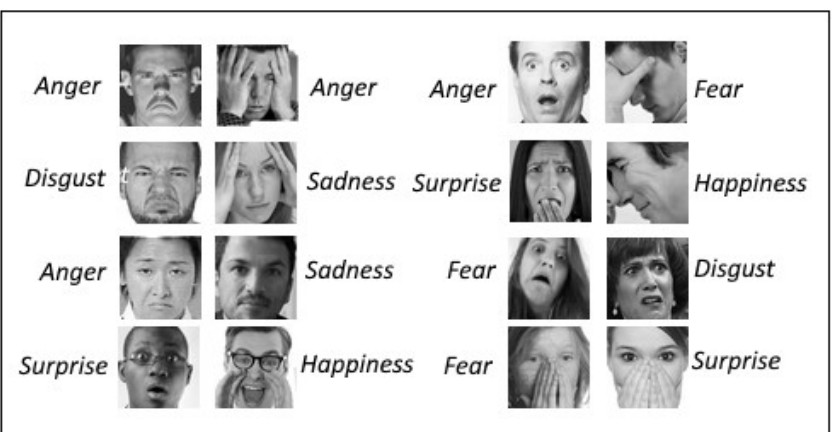

**Figure 4.** Some emotions from the Kaggle Dataset.

This presents some critical issues:

- Facial expressions, or gestures, are not always the primary characterization of images.
- The numerical distribution of classes is very uneven and, in some cases, limited (Tables 1 and 2).
- The "distance" between the reference classes is not very marked (we collected conflicting opinions of attribution through small surveys aimed at students).

**Table 1.** Training set.

| Training set | Total |
| --- | --- |
| L0 (Anger) | 3995 |
| L1 (Disgust) | 436 |
| L2 (Fear) | 4097 |
| L3 (Happines) | 7215 |
| L4 (Sad) | 4830 |
| L5 (surprise) | 3171 |

**Table 2.** Testset.

| Testset | Total |
| --- | --- |
| L0 (Anger) | 958 |
| L1 (Disgust) | 111 |
| L2 (Fear) | 1024 |
| L3 (Happines) | 1774 |
| L4 (Sad) | 1247 |
| L5 (surprise) | 831 |

The single images are grayscale and have a resolution of 48 × 48 pixels.

The training set has been appropriately relabeled by generating six different training sets from two classes ("reference emotion" and "other emotion").

In this scenario, it seemed helpful and not too expensive in computational terms to use SNNs (self-normalizing neural networks). These are networks robust to noise and disturbances and do not exhibit high variation in their training errors. In deep learning, a widely used technique is batch normalization, which leads to the normalization of neuron activation towards mean zero and unit variance. Each level is normalized and used as an input to the next level. SNNs, on the other hand, are self-normalizing, neural activations that automatically converge towards mean zero and unit variance. This property is ensured by the activation function, which consists of scaled exponential linear units (SELU). This characteristic accelerates convergence in the formation process. SELU learns faster and

better than other activation functions without needing further processing. The SELU activation function can be expressed mathematically as:

$$f(x) = \lambda \times x \text{ if } x > 0 \tag{1}$$

$$f(x) = \lambda \times \alpha \ (e^x - 1) \text{ if } x \leq 0$$

Graphically, it can be represented as in Figure 5.

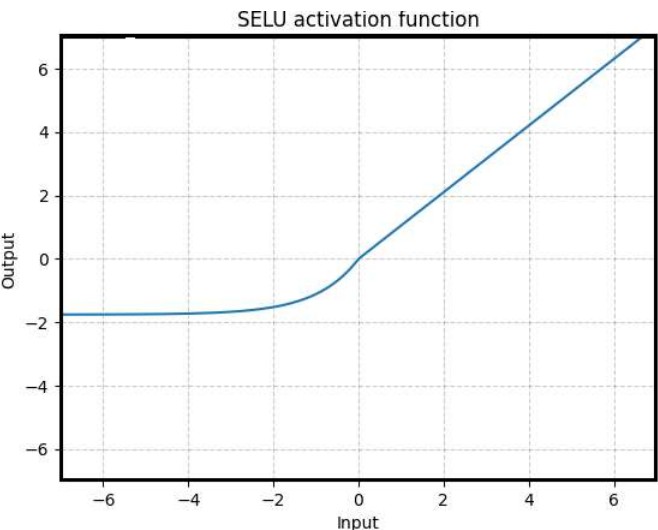

**Figure 5.** SELU activation function.

The "SELU" nonlinearity keeps the data standardized and prevents the gradients from becoming too small or too large. The effects are comparable to batch normalization while requiring significantly less calculation. In addition, the convergence property of SNNs towards mean zero and unit variance allows the training of deep networks with many levels and makes learning highly robust.

The configuration of the networks (Figure 6) is as follows:

- Input layer (48 × 8 × 1).
- Layer with linear activation function (105 neurons on average).
- Dropout layer, with a 30% activation rate; a layer present only during training that helps prevent the phenomenon of overfitting.
- Layer with SELU activation function
- A linear output layer (Figure 7), this layer has two outputs for individual neural networks.
  - Epochs = 60
  - Learning rate = 0.0001
  - Accuracy = 98.4%

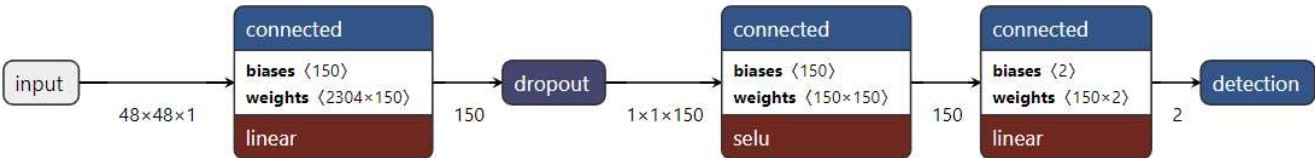

**Figure 6.** Network configuration.

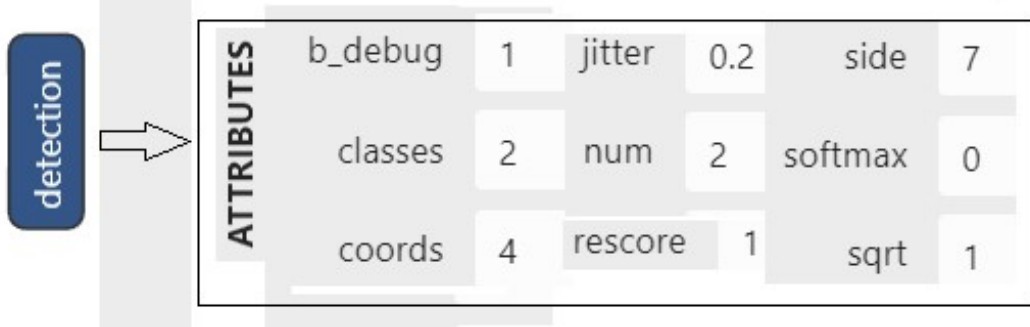

**Figure 7.** Attributes of output layer.

The control neural network has six outputs (as many as the emotions to be classified).

For the training, we used the Adam stochastic optimizer [50], with a learning rate of 0.0001 and a minibatch size of 200. This optimization algorithm can be used instead of the classic stochastic gradient descent procedure to iteratively update the network weights based on training data. Regarding loss function, we used multinomial logistic loss.

In addition, we excluded a certain number of images from the training set to make it as homogenous as possible, as they would have produced a misclassification due to their similarity to other emotions.

The test set was not altered, thus introducing an additional "stress" factor in evaluating the results.

We also used Netflix's Vectorflow Framework (Apache License Version 2.0), a minimalist neural network library for a single-machine environment written in D optimized for sparse data and low latency. D is a development language that can compile natively into many hardware and operating systems while retaining the simplicity of development like Python.

The proposed methodology is ensemble learning in a broad sense. Various ensemble techniques have long been considered and applied to AI network models; however, in these configurations, the models are complete and autonomous and could also be used outside the ensemble configuration. In our case, however, the single models (the single neural networks) can identify only one specific emotion. Therefore, more than a single network would be required to address the problem under examination. With our approach, the ensemble process is "distributed" between the homonymous layer and the networks, which, on the one hand, operate a classification for each emotion and, on the other, abstract information of the six classes that is then analyzed by the terminal layer for precise identification of the emotion.

For the Ensemble algorithm used, we applied the following rules:

- If multiple networks attribute different emotions to the same image, the network wins where the difference between the two outputs is higher (less uncertainty);
- If only one network performs the classification, no further investigation is carried out, which becomes the solution for the entire model;
- If no network classifies the input image as a specific emotion (all outputs are: ≪other emotion≫), then the ensemble layer will choose the emotion associated with the network for which the two outputs have a shorter distance between all other networks (greater uncertainty in classifying it as another emotion).

From a methodological point of view, a "control" neural network was implemented. A comparison was performed for the results obtained with the Ensemble AI architecture. This network is the same type as the individual SNN networks but built (six outputs) and implemented to classify the six emotions autonomously. The training was carried out with the same training set. Its configuration is the same as that used for the classifier (Figure 6).

## 6. Results and Discussion

In the experiments, we used a result validation approach through the control network. In Table 3 we reported the results for the six emotions analyzed. Success rates were achieved using the original test dataset and a training dataset reduced by several images to make it more consistent. The percentages refer to each type of image (emotion) in the case of the control neural network, to the single AI NN units, and to the entire model in the Ensemble algorithm.

**Table 3.** Success Rate.

|                 | Control Network | Sub Network | Ensamble |
|-----------------|-----------------|-------------|----------|
| L0 (Anger)      | 22%             | 62%         | 78%      |
| L1 (Disgust)    | 10%             | 18%         | 22%      |
| L2 (Fear)       | 82%             | 94%         | 95%      |
| L3 (Happiness)  | 43%             | 76%         | 81%      |
| L4 (Sad)        | 64%             | 83%         | 85%      |
| L5 (surprise)   | 54%             | 77%         | 80%      |

From the analysis of the data, there is a low value for the "Disgust" emotion for all types of networks used, both due to the low number of samples present in the datasets and the difficulty of its identification due to the lack of expressive separability from the "Anger" class.

We also found that network performance improved by moving from the single control network trained for the six types of emotions to those of individual networks built to identify a specific emotion (no false positives were detected). Moreover, finally adding the rules of decision and unification of the ensemble layer, we noticed that the performance of the network improved further. The success rates are almost all around 80%, with a peak of 95% for the "Fear" emotion.

We compared the model's performance with the other proposals in the related work (Table 4). Depending on the specific functionalities of the emotion recognition system, researchers used different methodologies, technologies and databases. In our analysis, we represented different experiences testifying to a varied scenario.

The results obtained show that the algorithm adopted ensures that facial emotion recognition results are compatible from the point of view of accuracy with the state-of-the-art.

The efficiency of the model was based on the type of network used. SNN networks allowed us to create a model with a reduced number of levels compared to existing models. This choice arose from some considerations related to the intrinsic quality of the model (e.g., not requiring any normalization of the input data) and the peculiarities of the dataset used in conjunction with the learning phase. The dataset is, in fact, not very homogeneous in terms of the numerical distribution of classes. Therefore, two of the qualities of SELUs are very valuable: it does not have a vanishing gradient problem, and neurons cannot die, as can happen with RELUs.

Then, the outputs of the individual networks are sent to the Ensemble layer, which has the task of improving the performance of the individual classification systems through the analysis of the results rendered.

We achieved 98.4% accuracy with a learning rate of 0.0001 and 60 epochs, while in the test phase we obtained an accuracy value of 85%, excluding the emotion disgust.

**Table 4.** Comparison of face emotion recognition systems.

| Authors | Purpose | Technologies | Database | Efficiency |
|---------|---------|--------------|----------|------------|
| Jn et al. [12] | Direct diagnosis of disease | • Deep Transfer Learning DTL<br>• MatConvNet | VGG-Face | 93% accuracy |
| Jn et al. [13] | Improved facial emotion recognition using pseudo RGB-D | • RGB-D sensor<br>• Depth plus<br>• Generator<br>• Adversarial network | • Bosphorous 3d Face<br>• CASUA 3d Face<br>• Bu-3DFe | 97% SSIM (Similarity index for measuring image quality) |
| Ghosh et al. [14] | Improved facial emotion recognition using physiological signals | • RNN<br>• HRV sensor | ISEAR | 96% accuracy |
| Dahua et al. [17] | Improved facial emotion recognition using physiological signals | • t-SNE<br>• PCA<br>• SVR<br>• PSD | SEED dataset | $0.625 \pm 0.029$ CCC (Concordance correlation coefficient) |
| Kanio et al. [19] | Improved facial emotion recognition using MEMS, environmental an physiological signals in real-life environments | • CNN<br>• CNN-LSTM | EnvBodySens | 95% accuracy |
| Suraj et al. [22] | Pain or discomfort recognition in patients monitoring | • CNN<br>• Haar classifier | N.D. | 79.71% accuracy |
| Bibbo' et al. | Face emotion recognition | Ensemble SNN | Kaggle | 98.4% accuracy in training and 85% in test |

The emotion collection was carried out based on static images. The usefulness of a video would provide spatiotemporal information for expression dynamics captured in a video sequence [51]. The temporal information is accurate, allowing us to perform better. However, it involves significant differences in the characteristics extracted during the duration of the transition and in the specific characteristics of the expressions depending on the subject's physiognomy. Possible approaches to solving this type of problem are costly in terms of computational time and complexity. Therefore, they do not efficiently reduce time redundancies in extracted frames.

The dataset used, which is complex in itself, presents for some classes a limited number of samples. Numerous images are not sufficiently clear, and can lead to misinterpretation. This led us to reduce the number of samples per class to make the entire dataset homogeneous. In order to improve accuracy, the number of samples for each class can be increased in future work.

Therefore, we believe that the proposed solution, due to the reduced computational load and its structural simplicity, can be used in the monitoring of the elderly to support medical staff in the assessment of the health status of patients.

## 7. Conclusions

In this article, we have developed a facial expression recognition system that can help improve healthcare. Despite the results obtained with technological progress in the development of automatic emotion recognition systems, this technology, as observed from the review of the literature, is not widely used in the health system.

The solution we propose is based on the Ensemble AI model. The methodology applied is part of an ensemble learning area in which the models, in comparison, discriminate two classes, with one referring to the specific emotion for the network for which it was trained and the other regerring to "other emotion" class. The advantages obtained were:

"Almost observable states". It is possible to investigate and highlight which module caused errors. It operates a debugging similar to the case of deterministic algorithms.

"Modularity and parallelism". Individual modules can be trained on different workstations, at different times, without synchronization between parts. This feature allows one to independently develop different configurations and calibrations of the specific module research groups.

"Embedded application". The Vectorflow framework in D language allows the realization of the model even in embedded hardware on many platforms and operating systems.

The results show an increase in performance compared to the control neural network, confirming that the proposed system can recognize emotions with high precision. With this system, doctors and healthcare professionals can constantly monitor, as part of the broader human activity recognition system, the psychophysical conditions of patients, detecting malaise, pain and fatigue and taking appropriate actions as needed.

This solution can be seen as a component of smart healthcare centers.

In the future, we plan to extend the work by investigating the infrastructure on a less complex dataset and analyzing video sequences.

**Author Contributions:** L.B. and F.C. contributed to conception and design of study. L.B. called the methodology. L.B., M.V. and F.C. investigated. L.B. and F.C. developed the model. L.B. wrote original draft of the manuscript. L.B., M.V. and F.C. contributed to write and editing of the manuscript. All authors have read and agreed to the published version of the manuscript.

**Funding:** This work is supported by the Italian MIUR Project under GRANT PON Research and Innovation 2014–2020 Project Code C35E19000020001, AIM 1839112-1: Technologies for the living environment.

**Institutional Review Board Statement:** Not applicable.

**Informed Consent Statement:** Not applicable.

**Data Availability Statement:** Public dataset Kaggle.

**Conflicts of Interest:** The authors declare no conflict of interest.

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
