# Peer review of "Emotional Health Detection in HAR: New Approach Using Ensemble SNN"

_applsci, doi:10.3390/app13053259_

Round 1

Reviewer 1 Report

Reviewer’s Comments:

The manuscript “Face emotion recognition in Human activity recognition in healthcare” is a very interesting work. This paper presents the computer recognition of human activity is an important area of research in computer vision. Recognition of human activity (HAR) involves identifying human activities in real-life contexts. Recognition is difficult as human activities are complex and highly diversified. Recognition of human activity plays an important role in interpersonal interaction. Through data collection relating to the activities carried out, information on the person's identity, personality, and psychophysical state are acquired. Artificial Intelligence usually identifies activities through data collected using different sources. These can be wearable sensors, MEMS devices embedded in smartphones, cameras, or CCTV systems. This technology finds wide application in numerous fields such as healthcare, surveillance, tracking and monitoring of people, sports, or the study of human-machine interaction or the recognition of gestures, faces, and human-object interaction. While I believe this topic is of great interest to our readers, I think it needs major revision before it is ready for publication. So, I recommend this manuscript for publication with major revisions.

1. In this manuscript, the authors did not explain the importance of the Face emotion recognition in the introduction part. The authors should explain the importance of Face emotion recognition.

2) Title: The title of the manuscript is not impressive. It should be modified or rewritten it.

3) Correct the following statement “Face analysis makes it possible to acquire useful diagnostic information for medical personnel to evaluate patients' state of health. In our work, we develop a new face emotion recognition model (FER) applying six neural networks SNN (Self-normalizing Neural Network) in cascade with Ensemble layer to detect emotions in images”.

4) Keywords: The Face emotion recognition is missing in the keywords. So, modify the keywords.

5) Introduction part is not impressive. The references cited are very old. So, Improve it with some latest literature like 10.3390/molecules27196564, 10.3390/molecules27207129

6) The authors should explain the following statement with recent references, “Finally, with Frame three, we will get the maximum evaluation of EM2t3 far superior to the others EMit3”.

7) Add space between magnitude and unit. For example, in synthesis “21.96g” should be 21.96 g. Make the corrections throughout the manuscript regarding values and units.

8) The author should provide reason about this statement “In addition, we have excluded a certain number of images from the training set to make it as homogenous as possible”.

9. Comparison of the present results with other similar findings in the literature should be discussed in more detail. This is necessary in order to place this work together with other work in the field and to give more credibility to the present results.

10) Conclusion part is very long. Make it brief and improve by adding the results of your studies.

11) There are many grammatic mistakes. Improve the English grammar of the manuscript.

Reviewer 2 Report

The authors must carefully revise the paper according to the comments.

1. In Section 1, the key challenges faced by deep learning based face recognition should be deeply discussed and analyzed.

2. The technical details of many key algorithms in Section 4 should be added.

3. The reasoning efficiency of the proposed method is encouraged to discuss.

4. More data examples in the dataset should be shown in Section 5.

5. The specific structures of Figure 2 and Figure 3 should be introduced in detail. Why three networks are used in each Framet?

6. The activation function and the final objective function of each layer of the model should be introduced.

7. The hyperparameter setting and training process of the model should not be ignored.

8. Why L1 (Disgust) performs worst in Table 5?

9. The summary and description of related work in the field are insufficient. The following related work of deep learning must be cited and discussed, including “VoxelEmbed: 3D instance segmentation and tracking with voxel embedding based deep learning.” International Workshop on Machine Learning in Medical Imaging. Springer, Cham, 2021: 437-446. “Compound figure separation of biomedical images with side loss.” Deep Generative Models, and Data Augmentation, Labelling, and Imperfections. Springer, Cham, 2021. 173-183. "Deep Facial Diagnosis: Deep Transfer Learning From Face Recognition to Facial Diagnosis," in IEEE Access, vol. 8, pp. 123649-123661, 2020. "Pseudo RGB-D Face Recognition," in IEEE Sensors Journal, vol. 22, no. 22, pp. 21780-21794, 15 Nov.15, 2022. “Improvement of generalization ability of deep CNN via implicit regularization in two-stage training process,” IEEE Access, vol. 6, pp. 15844-15869, 2018. 

Reviewer 3 Report

The author should improve the quality of the manuscript with the support of appropriate results. This work is lagging behind with adequate and supportive results. 

Reviewer 4 Report

Authors have developed FER methodology based on ensemble formulation. They incorporated AI.

The current version is not written up to the mark.

a) The abstract is totally verbose. It contains many redundant sentences.

b) Introduction is not well defined. The objective of the study is missing completely.

c) Methodology has also been represented in a very unorganized manner.

d) Literature review should come in late part of the introduction.

e) The images are in very bad shape

f)  The formulation is not at all well-defined.

g) The results and discussion part are too short to grasp what authors did actually.

Round 2

Reviewer 2 Report

All the comments have been well revised and thus this paper can be accepted for publication.

Reviewer 3 Report

No comments